**www.cambridge.org/ext**

extinction crisis; bird trade; biodiversity conservation; wildlife trade; captive breeding

**Corresponding author:**
Sicily Fiennes;
Email: sicilyfiennes@gmail.com

# Rethinking extinction "crises": The case of Asian songbird trade

Sicily Fiennes[1] , Novi Hardianto[2], Silvi Dwi Ansari[2], Asri A. Dwiyahreni[3], Tom Jackson[4], George Holmes[5], Christopher Birchall[6] and Christopher Hassall[1]

[1]School of Biology, Faculty of Biological Sciences, University of Leeds, Leeds, UK; [2]Yayasan Kausa Resiliensi Indonesia, Jakarta, Indonesia; [3]Research Centre for Climate Change, Faculty of Mathematics and Natural Sciences, Universitas Indonesia, Depok, Indonesia; [4]Numiko, Leeds, UK; [5]School of Earth and Environment, Faculty of Environment, University of Leeds, Leeds, UK and [6]School of Media and Communication, University of Leeds, Leeds, UK

## Abstract

Different stakeholders and actors frequently describe environmental challenges as 'crises'. These crises are often wicked problems that are difficult to resolve due to the complex and contradictory nature of the evidence and knowledge systems surrounding them. Here, we examine a crisis narrative surrounding the IUCN-declared Asian Songbird Crisis (ASC), with its epicentre in Indonesia, where an extensive birdkeeping culture persists. We investigate how bird extinction is perceived by different actors, particularly conservation law enforcement and practitioners working in this space. We unravel local perspectives on the complex relationship between bird trade and extinction through one-to-one interviews and focus groups. Our examination reveals a diversity of attitudes to the ASC, with many law enforcement actors not recognising the crisis label. Market mechanisms result in complex shifts in harvesting pressure onto one or more closely related similar species. The findings challenge the prevailing notion that species extinction significantly affects wildlife trades, emphasising the plastic nature of trade and the coming and going of species fashions. By revealing the divergent views of actors on extinction and the ASC, we highlight the need for shared language, particularly the implications of the 'crisis' label, around species extinction.

## Impact statement

This study explores the complex nature of the Asian Songbird Crisis (ASC) in Indonesia, focusing on different actors' perceptions, especially conservation law enforcement and conservationists. Through interviews and focus groups, the results uncover diverse attitudes toward the ASC, challenging the idea that species extinction significantly impacts wildlife trade. The findings highlight the dynamic nature of the trade and the shifting popularity of species. The study also reflects on the implications of labelling such environmental issues as 'crises' and underscores the complexity of addressing wicked problems in conservation.

## Introduction

### Understanding crises

In environmental sciences, crisis theory refers to a framework that examines how ecosystems or species respond to external pressures. If many species face a high risk of extinction within a relatively short period, this response is known as an extinction crisis (Pimm et al., 2014). Biological extinction, in the simplest sense, is defined as the point at which the population of a species is zero. More complex concepts describe species that are unlikely to avoid extinction as 'functionally extinct' or, where a species persists only in captivity, as 'extinct in the wild' (Roberts et al., 2023). Anthropogenic extinction crises are often the subject of scholars and practitioners within the discipline of conservation, which Mitchell (2016) describes as the 'systematic management of life forms to ensure their survival'. Conservation's intimate relationship with extinction is worth considering given that conservation itself has often been described as a crisis-oriented discipline (Soulé, 1985), faced with problems that demand 'immediate action' (Mitchell, 2016).

Crises themselves are difficult to define and often fraught with contrasting definitions. One school of thought sees crises as discrete events, such as the 2004 Indian Ocean tsunami or Hurricane Katrina in 2005 (Paglia, 2015). These time-limited events could also be short periods of political upheaval or economic shocks, such as the 2008 financial crisis. Balestrini et al. (2020) understand crises as the critical moment of any process that has arrived at a 'decisive juncture'. Ecology does not refer explicitly to crises, but these 'decisive junctures' manifest during regime

shifts as an ecosystem moves from one relatively stable state through a tipping point to another stable state from which it is difficult to return (Clements and Ozgul, 2018). These switches or oscillations are characterised by early warning signals as systems approach 'critical transitions' (Clements and Ozgul, 2018). These ecological early warning signals mirror the early warning phase defined in crisis management studies, which includes six phases that make up a crisis episode (Paglia, 2015).

Paglia (2015) and Kelz (2022) assert that crises at the forefront of conservation discourse, such as species extinction and climate change, cannot be 'proper' crises since they cannot be managed as discrete events where the status quo can be restored. Hence, dramatic ecological changes only become crises when we incorporate the broader sociopolitical implications of biodiversity loss rather than simply species decline and environmental change. For instance, marine fisheries experience population crashes or collapses but are only labelled as part of a "food crisis" when considering the anthropogenic causes and anthropocentric consequences. However, it is easy to see how such ecological transitions could become permanent, for instance, the transition from the past, where a species was alive, to a future in which it does not exist (Jørgensen, 2022).

In addition to the reversibility of critical transitions, their time-scale is also debated. Although he calls it a crisis, in his work on defining and introducing the term 'Capitalocene', Moore (2017) argues that the origins of the ecological crisis are found in the British-led Industrial Revolution, which marks a longer period of socioenvironmental change rather than a more discrete juncture. Ferdinand (2022) goes further and argues for a much earlier colonial fracture, foundational to modernity and globalisation since the 1500s. Such a long-term, chronic view of crises is also suggested by Roitman (2013) when analysing the 2008 financial crisis. Thus, crises are not merely logical, ontological assessments of periods of change but can also be political denunciations of situations spanning long periods (Koselleck and Richter, 2006). Here, we consider what happens when this crisis label is applied to conservation challenges and how that label shapes knowledge and politics.

The framework of a conservation crisis can depoliticise inherently political situations by overly focusing on wildlife and their ecology. Conservation practitioners must act 'before knowing all the facts' (Soulé, 1985), often without valuable sociopolitical facts which provide the broader context of a putative crisis. Classical definitions of extinction do not account for nonbiological (for instance cultural) corollaries accompanying species loss. Conservation has long faced calls for greater recognition, participation and inclusion from the Global South, including indigenous perspectives (Masse et al., 2020). Conservation biology and the more humanities-focussed field of extinction studies prejudice the biological and assumed empirical permanence of extinction, a sterile understanding that extinction is something that 'happens' to nature (O'Key, 2023). Societies with more connections to nature, often that hold indigenous worldviews, understand a more mutualistic relationship between humans and nature, where extinction is a cultural or even sociopolitical problem. For example, indigenous groups like the Anishinaabeg people observe that animals actively withdraw from the world when societies break human-animal treaties (O'Key, 2023). This cultural appreciation for human-nature interactions is in opposition to forms of 'new conservation' (Büscher and Fletcher, 2020) that rely on neoliberal frameworks to understand nature and species' roles in ecosystems as ecosystem services that monetise both extinction and conservation (O'Key, 2023). Elsewhere, there are narratives of intrinsic trade-offs between conservation and the prevention of extinction and the economic development agendas pursued by national governments (Otero et al., 2020).

## The Asian songbird crisis as a nested crisis

One example of a crisis narrative in conservation is the Asian Songbird Crisis (ASC). This crisis was labelled and declared by the International Union for the Conservation of Nature (IUCN) in 2017 in response to high rates of (il)legal bird trade across Southeast Asia (Shepherd and Cassey, 2017). According to the IUCN, this crisis declaration aims to prevent the 'imminent extinction of songbirds threatened by unsustainable trapping and the trade in wild-caught passerines' (Asian Songbird Trade Specialist Group, 2018). The declaration of this crisis was not triggered by the crossing of a particular tipping point but rather as the result of an ongoing process where 'the degree of pressure on songbirds in Asia is devastating and has long been grossly underestimated' (European Association of Zoos and Aquaria, 2023). A range of actors, largely in the Global North, such as zoos, universities and conservation organisations, have united for 'the common purpose of Asian songbird conservation'.

The global hotspot of bird trade is in Indonesia, where many practices involve birds for use as pets, food, prayer release or as omens (Gilbert et al., 2012; Su et al., 2014; Marshall et al., 2019). The wild bird trade in Southeast Asia, including Indonesia, is a somewhat atypical case study because illegal trade happens in the 'open', where protected species are displayed openly in marketplaces or sold in violation of the national quota system (Shepherd et al., 2016; Shepherd and Leupen, 2021).

If we accept the ASC as a (singular) crisis, it is also nested within a larger global ecological crisis, forming a hierarchy of 'nested crises' (Kaukko et al., 2021). Some claim the legal and illegal trade in wildlife has created an 'extinction market', where species are traded until they become extinct (Felbab-Brown, 2017). The extent to which wildlife trade might drive a potential songbird extinction crisis is unclear, especially since many species threatened by trade face the synergistic threats of habitat fragmentation and climate change (Sagar et al., 2021). Hinsley et al.'s (2023) global review on wildlife trade-driven species extinction found 294 reports of 'successful' extinctions. Thirty-four reports related to wild birds, though not yet from the songbird trade (Hinsley et al., 2023).

Nonetheless, since its declaration in 2017, the ASC has arguably intensified as a 'thing' (pressure on birds) and a 'crisis' (a discourse). Several species have been uplisted in conservation assessments by the IUCN, and 63 (sub)species are listed as conservation priorities by the IUCN Asian Songbird Trade Specialist Group (ASTSG, 2022). Conservationists, largely from Global North NGOs and universities, have dominated ASC discussions, framing this issue as a cultural and legal problem that stems from demands for birds and inefficient law enforcement. In contrast, in 2018 Indonesian birdkeepers lobbied to downlist popular species, such as the White-rumped Shama, *Copsychus malabaricus* (locally extinct in most of its Indonesian range) (Gokkon, 2018b). The birdkeepers' rationale was that these species were bred on a large scale and thus far from endangered, and their uplisting and effective banning would have uneven socioeconomic impacts across the songbird economy (Gokkon, 2018b). These conflicting views typify the tensions between groups of actors in the landscape of (conservation) crises.

To grasp the local implications of global or regional 'crises', we explore the ASC in Indonesia as a case study of how extinction

crises manifest. Our goal is not to deny the severe impact of the bird trade on wild populations but rather to examine the implications and repercussions of declaring it a crisis. The ASC and its designated task force, the ASTSG, of which SF (the primary author) is a member, operate under the IUCN Species Survival Commission (SSC). Here, we explore how various local and international actors more embedded in power structures, such as conservationists, law enforcement and practitioners, perceive bird extinction within the ASC narrative.

## Methods

We conducted field observations, semi-structured interviews and focus groups with law enforcement officers, conservation community members, environmental NGOs and academics primarily based in Indonesia. The questions asked during interviews and focus groups are part of wider discussions on the ASC (see the Supplementary Materials). Data collection took place in cities within Java (Jakarta, Bandung, Yogyakarta and Surabaya) and West Kalimantan (Pontianak), renowned for their role in the bird trade (Chng et al., 2015; Chng et al., 2016; Chng and Eaton, 2016; Rentschlar et al., 2018). Although Pontianak is not the largest urban market outside Java, recent research indicates it is an emerging bird source, with trade levels comparable to neighbouring islands (Rentschlar et al., 2018).

### Field observations

We visited nine bird marketplaces (one in Bandung, Yogyakarta and Surabaya; three in Jakarta; and collections of shops in Pontianak) between January and June 2023. During each marketplace visit, we observed variations in the physical architecture of marketplaces, microscale dynamics between traders and buyers and aspects of the birds' health and welfare. Marketplace visits were approximately two hours long, with more time spent at larger marketplaces. All marketplaces were visited at least twice, apart from Jatinegara market in Jakarta, which was seen only once.

### Interviews

We purposively selected participants engaged in bird trade issues from conservation organisations and universities in Indonesia and abroad. We conducted 16 semi-structured interviews in May and June 2023 (half in person and half conducted online via Zoom). Participants were asked about the consequences of species extinction on bird trade and birdkeeping practices.

### Focus groups

We chose focus groups as they offer flexibility and openness, crucial for discussing sensitive topics (Newing et al., 2011, p. 54). Focus group discussions (57 participants in total, ranging between 8 – 13 per discussion) were conducted in the five cities, with focus group participants (FGPs) including representatives from agencies such as the Indonesian National Police (POLRI), the nature conservation agency (BKSDA), environmental law enforcement (GAKKUM) and agricultural quarantine. We obtained verbal permissions from agency heads through Yayasan Kausa Resiliensi, the local NGO with which we collaborated. These discussions explored whether law enforcement sees its job as preventing extinction (a more ultimate goal) than enforcing a law (a more proximate goal).

### Ethical considerations

Participants provided their free, prior, and informed consent (FPIC), and data from interviews and focus groups were anonymised (Ibbett and Brittain, 2020). Participants were identified by codes based on their international (I) or national status (L(ocal)), and sector (AC (ademic)), L(aw E(nforcement)), NG(O)). For instance, "AC_I_01" represents the initial participant in the international academic category, while all LE were local, so they are denoted as LE_01 and so on. Our methodologies received ethics committee approval from the University of Leeds (AREA FREC 2023-0419-521) and were conducted under a research permit (311/SIP/IV/FR/11/2022) from Indonesia's National Research and Innovation Agency.

### Data analysis

The data analysis followed a two-stage process. Initially, we conducted a discourse analysis to examine framings, narratives and language around songbird extinction. Subsequently, we performed thematic analysis, employing inductive coding following the method by Newing et al. (2011, p. 6), exploring themes around bird extinction, sustainable trade and their combined consequences, using grounded theory principles. Interviews were primarily conducted in English and later transcribed. Focus groups, conducted in Indonesian, were recorded and transcribed by student assistants and later translated using online tools and support from the research team.

## Results and discussion

Below, we combine multiple lines of evidence drawn from interviews with NGOs and academics, focus groups with law enforcement professionals and visits to a wide range of bird marketplaces to extract four key themes concerning the ASC at a local scale in Indonesia. Firstly, we find that the 'crisis' framing of the ASC does not translate from the scale of global conservation to the day-to-day realities of conservation authorities on the ground. There are tensions between the absolutism of crisis and the complex and conflicting realities presented by local actors. Secondly, local actors see difficult links between species decline, extinction and the trade chain's responses. Thirdly, species extinction has limited economic consequences for the trade chain but may affect the ecosystems of harvested areas. Finally, we examine the perceptions of our informants towards birdkeepers and captive breeding as solutions to anticipated extinction.

### Perceptions of crisis

Our data shows that Indonesian government agencies responsible for regulating the songbird trade often did not perceive there to be a crisis. In workshops, many participants asked us, '*What crisis*?' This difference in perception of songbird conservation amongst law enforcement officials may be attributed to their focus on day-to-day operations and short-term, local trends. This viewpoint might also result from different education and training for law enforcement around the bird trade compared to conservation NGOs. For example, their work involves extensive protected species lists, '*in our area, there are around 500 protected animals [birds], and we do not remember all of them*' (LE_17).

Consumers and law enforcement alike may overlook the threat of extinction due to the widespread availability of at-risk species in marketplaces. For example, one interviewee remarked, '*If they do*

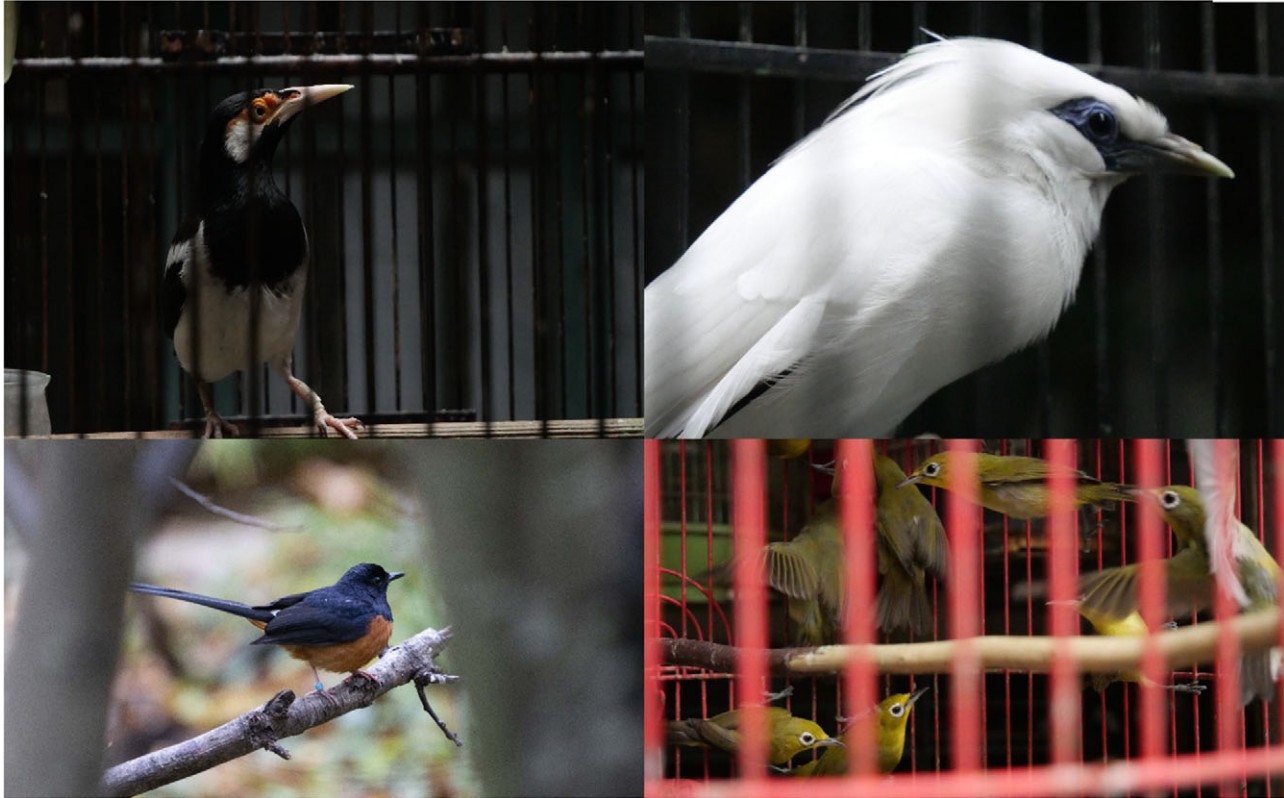

**Figure 1.** The plate depicts a Javan Pied Starling, *Gracupica jalla* (top left), Bali Myna, *Leucospar rothschildi* (top right), White-rumped Shama, *Copsychus malabaricus* (bottom left), Lemon-bellied White-eye, *Zosterops chloris* (bottom right). All photos by Sicily Fiennes.

*not have…any conservation background, nobody will suspect the White-rumped Shama will be protected because they are just so numerous in the market'* (NG_L_4). In Indonesia, the bird trade is a 'grey market,' where illegal items are passed off as legal (Dickinson, 2022). Species such as the White-rumped Shama, *Copsychus malabaricus* (Figure 1) can be passed off as 'grey' if wild individuals are mislabelled as captive-bred and vice versa using the standard numbered rings used to track captive-bred birds. The fact that consumers encounter both wild and captive-bred birds in marketplaces contributes to distorted perceptions of extinction. Courchamp et al. (2018) showed that marketing charismatic species led people to believe these species are less endangered.

Materiality matters beyond a bird's source, as evidenced by uneven research attention and funding to high-profile, charismatic species such as tigers, elephants and rhinoceroses (Challender et al., 2014). Dickinson (2022) calls this 'fleshy geopolitics', where a wildlife product's fleshy material properties shape markets, regulatory systems and corruption. Bird trade is facilitated because birds are small and relatively easily transported, with one species fungible for another. As one participant phrased, *'more attention will be given to saving a single tiger than 1,500 songbirds that people do not even know their names and think there's already a lot in the market'* (AC_L_1), revealing a gap between conservationists' messages and local experiences.

## Confirming extinctions

Our findings also reveal the challenges of declaring bird extinction that are shared by local actors and the international conservation movement. An example of the uncertainty over formal declarations of extinction is the case of the Javan Pied Starling (Figure 1) (Baveja

et al., 2021), which has not been formally declared extinct by the conservation community but is recognised to be *'no longer really in the wild'* (AC_I_1).

There needed to be more consensus on the extent to which bird trade can lead directly to extinctions. One interviewee emphasised the role of trade as *'the extinction driver in Asia right now'*, commenting that *'I have seen a bunch of species get depleted in numbers: various bulbuls…hill mynas…most of them are going extinct because of the bird trade'* (AC_I_1). On the other hand, another interviewee commented that *'it is very difficult for things [birds] to become extinct'* (NG_I_2). In contrast, law enforcement hypothesised that *'for extinct species, the demand in the market will be higher…and the value of the birds will also increase'* (LE_14).

Others from NGOs did not support this interpretation of low substitutability. Several participants described how when one species declines, trappers *'[switch] to another species'* (AC_I_1)*, seek another bird'* (AC_L_2), or *'[trappers] just substitute'* (NG_I_2). While the participants did not mention it explicitly, these comments resemble the process of 'substitution elasticity' (Fraser et al., 2023). Understanding how substitutions occur is critical when considering why so few extinctions have resulted from the songbird trade (Hinsley et al., 2023). Interviewees suggested that, at a certain point, the efforts of finding rarer species do not outweigh the benefits of substituting for another species. This shift may be of no consequence; *'I do not think anyone is that bothered if a species goes extinct. There will be another one that takes its place'* (NG_I_2).

Substitutions can occur within a species complex, between species in the same genus, or based on shared traits within the same family. For example, some species have enough variation that they are split into several subspecies, like *'the White-rumped Shama,*

which is already very rare in Indonesia [at the subspecies level], but they import it from outside: Malaysia, Thailand, the Philippines' (NG_L_6). Species in genera with cryptic diversity, where multiple species look very similar, are vulnerable to 'true' substitutions within the same genus. The *Zosterops* White-eye genus is a common example; *'take the Sangkar White-eye, absolutely millions of them in trade, and then they become exceedingly difficult to find, and then all of a sudden, the market's flooded with Lemon-bellied White-eyes'* (NG_L_2). In some cases, trade pressure may be displaced to species within the same family with similar traits, such as colour, song and size. In the bulbul (Pycnonotidae) family, 'sound-alike species' are also at risk; with the *'straw-headed bulbuls [Figure 1]… suddenly, they start using grey-cheeked bulbuls instead, which is the second-best choice if you are into bulbul songs, so [grey-cheeked bulbuls] become rarer in the field'* (AC_I_1).

Others assert that generally, demand is never species-specific, that it is *'for the traits, or the use of the species…[there] is always going to be something else filling it in'* (AC_L_1). This 'filling in' of species has been described as a continuous process of 'rolling local extinctions' (Jepson and Ladle, 2005; see also Jepson, 2010) or 'near extinctions' (Eaton et al., 2015), which drive species substitutions and an increase in trapping effort, even though *'there are still birds everywhere. People go further afield'* (NG_I_2). This process also has permanency; *'if we talk about Indonesia, the trade will never stop because the tendency of traders will not let trade disappear'* (NG_L_6).

## Socioecological impacts of trade

A common theme when discussing the consequences of trade on the wider environment was the ecological and socioeconomic consequences of species loss. Broadly, indiscriminate and opportunistic trade can reduce the diversity of a species community, leading to a reduction in community function; *'it will be that the environment or ecosystem equilibrium is disturbed'* (AC_L_2). This idea was echoed in a focus group: *'the loss of one component of the ecosystem will disrupt the balance'* (LE_25).

Trade could disrupt ecosystem stability, for instance, *'food chains, pollination of seeds, and the fertilisation of the soil by droppings'* (LE_5). Particularly, *'[a bird's] specific function might be lost, for example, an insectivore, or nectarivore'* (AC_L_2). One FGP gave a specific example; *'[a decrease in] owls…would mean an increase in the population of rats and snakes'* (LE_29). Interviewees also linked ecosystem instability to the loss of habitat specialists and other indirect threats of deforestation; *'I think there will be many species that disappear… that are dependent on forests areas…the number of birds that you can harvest has decreased because there's no more habitat…they do not live in oil palm plantations and other agricultural areas'* (AC_I_1).

The literature would suggest that birds have important roles in community ecosystems. However, the recognition or understanding of these roles amongst trade actors is unknown, with one interviewee commenting that these roles are *'completely unknown and misunderstood by the [birdkeeping] community'* (NG_L_7). The ecological impacts that result from the loss of rare species, at present the concern of the ASC, are also unclear. Kleijn et al. (2015) argue that common species (particularly bees) handle most pollination tasks, making the loss of rare species less economically significant. Still, some rare species may have unique roles or indirect contributions within their ecosystems, and geographically restricted but locally abundant rare species can still provide valuable services (Dee et al., 2019).

Common species are also at risk, often sold in larger numbers and worse conditions due to their perceived low value and unprotected status. In a Jakarta bird marketplace, for instance, SF (first author) observed live suffocating several Scaly-breasted Munias, *Lonchura punctulata*, and a dead individual in a water pot. An interviewee was *'watching a cage full of munias being spray painted… in the first five minutes, 20% just dropped dead…I assumed from the digestion of the chemicals'* (NG_I_2). A few interviewees also emphasised that *'harvesting unprotected species is still harmful'* (NG_L_7).

In Indonesia's songbird economy, the socioeconomic consequences of population crashes may *'affect people who rely on it for their livelihoods and jobs'* (LE_41). Conversely, an NGO staff member suggested that population crashes will *'only really bother high-end traders…and those in the field trapping the birds…there are fewer things for them to trap and less money'* (NG_I_2). This may also privilege wealthier consumers: *'those who can own and sell birds [will be] people who can afford to pay a high price'* (LE_27).

In the long term, as rarer species decline, both the songbird economy and the composition of ecosystems in Indonesia may become dominated by resilient, common species. According to Hughes et al. (2022), there has already been a loss of morphological diversity among birds in East Asia. As stated by one interviewee, *'You might come to a point where we are going to have to live with the resilient birds'* (AC_L_1). While specific data on longer-term evolutionary consequences of harvesting on songbirds is lacking, research on bighorn sheep suggests that intense harvesting of certain traits has long-term consequences, for instance, decreased horn size in offspring (Rice et al., 2022).

## Views on the role of breeders and breeding

In response to the problem of trade-induced declines, many interviewees added to the discourse on the role of birdkeepers in songbird conservation, birdkeepers' savourism towards wild bird populations, and the role of captive breeding in mitigating the effects of wild songbird harvest. Similarly, we consider perspectives on how the state (intervenors) perceive the people being intervened on (birdkeepers) through deploying the crisis narrative.

Participants described how birdkeepers perpetuate notions of saviourism, relating to heroism and resistance, where they prevent extinctions and alleviate marketplace conditions. This savourism directly opposes racialised stereotypes in earlier conservation commentaries, which describe aviculturists who demand rare birds as malevolent (or ignorant) (van Balen et al., 2000). In contradiction to these stereotypes, the heroic saviour says that *'by buying the birds, they are also saving them from extinction'* (NG_L_5). Breeders of rare birds may also believe that *'they are helping the government to conserve the birds'* (NG_L_5). Birdkeepers may also go *'to the market and see a certain animal in a condition they believe deserves to be helped…so a sense of compassion arises'* (NG_L_6).

Sections of the Indonesian government also conceptualise breeding as conservation. For instance, official governmental guidance states that registered breeding facilities can catch protected species in the wild and sell the offspring, then release 10 % of their captive-born stock back into the wild (Gokkon, 2018a). One FGP referenced this policy, stating that *'the government should prioritise captive breeding (for protected animals) or animal husbandry (for unprotected animals). Bird breeding in Indonesia is widespread, and 10% must be returned to nature* (LE_5). This pro-breeding approach aligns with the current Indonesian President, Joko

                                                                                                        

Widodo, who supports captive breeding (Cabinet Secretariat of the Republic of Indonesia, 2018; Nuruliawati, 2018). In 2018, at a singing competition held at his residence in the Bogor Botanical Garden, he stated that he saw captive breeding as a 'space for bird enthusiasts that can also prevent birds from extinction' (Cabinet Secretariat of the Republic of Indonesia, 2018).

In contrast, the resistant saviours see themselves in place of state-sponsored conservation. This attitude appears to be linked to a mistrust in the government, where '*keepers think that if they keep the birds, they are also conserving the birds from extinction because they think that our forests are getting lost* (NG_L_5), and the perception that birds are safer in their care: '*birdkeepers think it is better to keep the bird in my cage, so I still enjoy the song and can see the birds*' (NG_L_5). This attitude shares similarities with civil society resistance observed in urban areas of the Global North, such as Northern Europe and eastern parts of the US, where citizens engage in acts like 'guerilla gardening' or 'citizen greening' in response to a perceived lack of environmental stewardship (Baudry, 2014). It is possible that some birdkeepers still want to avoid state and institutional involvement, and their individual breeding efforts can be conceptualised as guerilla conservation

The extent to which captive breeding can be a conservation solution was debated; one interviewee suggested it '*is the only realistic solution at the minute*' (NG_I_1), whereas others were less positive. One participant expressed concern that '*there are not the breeding facilities to develop or elevate the number of the individuals of the species*' (AC_L_2) and '*there's not enough availability in the market. [Captive bred birds are] too expensive…beginners…they will start with a wild one because it is easy and cheap* (AC_L_1). Further, some participants felt that the financial rewards of captive breeding of near-extinct species are prioritised over conservation gains: '*[many] people [try] to breed a species they like, such as the green leafbird…[they] might think, if a species is heading towards extinction when I succeed in breeding it, the price will also be high*' (NG_L_5).

### Alternative framings to the ASC

Specific framings of trade-induced bird declines can drive responses and financial support. To illustrate, labelling the illegal hunting of elephants as a 'poaching crisis' led to strict, culturally specific responses, including shoot-to-kill policies against poachers in South African national parks (Lunstrum, 2014). Fletcher (2015) highlights the 'ecotourist gaze', representing a desire to connect with nature as an escape from industrial civilisation. Framing the Southeast Asian bird trade from outside of Indonesia, based on conservation ideologies, resembles Fletcher's ecotourist gaze. The crisis narrative itself relies on conservation mechanisms rather than the integration of birdkeepers and broader civil society.

Thus, the ASC takes the 'conservationist gaze' by privileging wildlife conservation over the needs of humans who benefit from songbird trade or even trade actors worried about bird declines, exemplified by the Silent Forest Campaign (European Association of Zoos and Aquaria, 2023).

Reminiscent of Rachel Carson's 'Silent Spring' (Carson, 1962), a silent forest erases human inhabitants of forests and reifies colonial notions of 'untouched' nature (Ferdinand, 2022). The conservationist gaze focuses on rare species, making nonthreatened species the 'unloved subjects of extinction' (Mitchell, 2016). Similarly, the protected-unprotected binary reinforced by law

enforcement prejudices commoner species. Framing a problem as a crisis attracts more funding, as powerful images inspire political mobilisation, and media visibility plays a crucial role in establishing crises (Väliverronen and Hellsten, 2002; Kelz, 2022). This is the case with the ASC, which post-declaration has received greater attention in academic research (Mirin and Klinck, 2021) and increased international funding for songbird conservation.

Roitman and others reveal that crisis narratives tend to advocate a return to a restricted status quo, hindering fundamental change. The use of crisis language narrows our focus, obscuring capitalist systems of oppression and exploitation (Khasnabish, 2014). It is important to acknowledge that the unsustainable harvest of songbirds is partly driven by rising unemployment and economic necessity (Lucas, 2011). Notably, law enforcement was concerned that '*Indonesia's economic needs are still very high. Birds have a significant market value*' (LE_5). (Il)legal bird trade may continue due to unaddressed global structures of inequality that drive illegal wildlife trade (Duffy, 2022, p. 30).

As a conservation intervention, the ASC lacks support from Indonesian agencies and songbird trade participants, neglecting long-term strategies (Masse et al., 2020). Conservation interventions without complete stakeholder buy-in often fail (Cooney et al., 2021), and the declaration of the ASC may falter as the impact on wild bird populations worsens without necessary local adaptations that incorporate the lived experiences of actors such as trappers and hunters. In work on attitudes towards IWT, Arroyave et al. (2023) found that rural communities justified wildlife utilisation since they are deprived of other production means and wildlife is used to meet basic needs. Similarly, one participant mirrored this necessity '*whenever I go into really rural areas, like the West Sumatra islands, I find the local police to be completely unreceptive to any arguments of having to jail…or…apprehend someone who is involved in the bird trade…they will just let them do whatever they want to do, because these are just people following their livelihood, trying to make a living, trying to feed their families*' (AC_I_1).

In addition to our suggestion to broaden the range of actors engaged in songbird conservation, here we present an alternative, framing bird trade within an unsustainable trade narrative; 'The Asian Songbird Challenge'. Following Roitman, we do not suggest that there is a singular alternative to crisis-framed narratives (Khasnabish, 2014). However, this framing is based on the similarities between the bird trade and the fishing industry. Though overfishing and trapping wild birds in Southeast Asia do not perfectly equate, the rapid population declines leading to 'fishing down the food web' (Pauly and Palomares, 2005), quotas, and shifts in focus to cheaper species when stocks collapse are commonalities. Although ecosystem service narratives have been criticised as neoliberal (Silvertown, 2015), the emphasis by law enforcement on the functional roles of birds and the ecosystem-level impacts of bird trade may be a more useful message for communicating the challenges of songbird trade governance rather than through a conservation crisis. Our findings suggest, therefore, that a more effective and inclusive framing would weave together biodiversity and ecology with a greater focus on socioeconomic factors that are most relevant to local actors.

Looking at environmental humanities scholarship, scholars such as Jørgensen (2022) argue that 'extinction [as a process] is not a linear phenomenon', offering a less final and more optimistic perspective, in contrast to the catastrophism that extinction crisis discourses offer. This challenge to the linearity of extinction is

inspired by a diverse range of observations, including indigenous knowledge systems that consider extinction as a change in interspecific relationships rather than a final death biological extinction, the survival of biologically extinct species in cultural artefacts, the evolutionary regeneration of lost life forms, and the "rediscovery" of those species that were never really lost (for instance, species rediscovery rates are at an all-time high (Scheffers et al., 2011)). Likewise, the ASTSG acknowledges that 'the growing threat to an ever-increasing number of songbird species can be reversed'. One interviewee also highlighted the potential for this reversal and species recovery: *'I think many species are thriving better than we thought they were'* (AC_L_1). There are parallels with discussions on fisheries and population decline, where shifts are seen as crashes or collapses but not as complex, multidisciplinary crises or discussions on global extinction. This potential for reversal of trends for declining populations of songbirds is similar to 'recovering stocks' when fishing pressure is reduced for commercially important species. Thus, reinterpreting extinction as Jørgensen does, by 'anticipating extinction' or allowing populations to recover, allows us to shift to more interdisciplinary and multifaceted understandings of species decline that are separate from traditional Western scholarship definitions.

## Conclusions and recommendations

The crisis narrative obscures aspects of the Asian songbird trade, including state actors not recognising a crisis. Bird trade is a 'daily wildlife trade', particularly in Indonesia (Duffy, 2022, p. 35). Despite few official extinctions resulting from the ASC, the Southeast Asian bird trade challenges simplistic extinction narratives due to substitution elasticity and changing traded species. We recommend using the 'Asian Songbird Challenge' for a nuanced perspective, avoiding oft-hyperbolic language such as 'crisis' or 'silent forest'. Drawing inspiration from overfishing narratives, we acknowledge the severity of wild bird extraction with a framing distinct from the conservationist's gaze. While an analogy to the fishing industry is needed, wildlife trade research must improve upon the industry's failures (Crona et al., 2019). We do not argue that a simple change of terminology will resolve the issue of a lack of agreement on the situation. However, framing songbird trade as a challenge with many components - singing contests, breeding programmes, sustainable harvesting - could weave together the cultural and ecological priorities of the different actors, as presented here.

Building on earlier, unheeded calls from researchers such as Jepson (2010), we propose tangible cooperation among governments, conservation practitioners and birdkeepers, bridging gaps for sustainable futures. Beyond the Bali Myna's success, more incentives must be given for breeder integration into conservation efforts. Dialogues are needed for sustainable Southeast Asian bird trade, utilising knowledge and infrastructure from birdkeepers and conservation breeding centres (Marshall et al., 2019). Our findings aim to stimulate ongoing dialogues and promote captive breeding as a valid tool in songbird conservation.

**Open peer review.** To view the open peer review materials for this article, please visit http://doi.org/10.1017/ext.2024.20.

**Data availability statement.** All data in this study have been anonymised and are unavailable for public access.

**Acknowledgements.** The authors would like to thank Jonathan Roberts, Valentina Fiasco, Timothy Brown, Karlina Indraswari and James Eaton for their comments on the manuscript. We would also like to acknowledge the valuable work of student research assistants during data collection: Luthfiyyah Damayani, Raden Nicosius Liontino Alieser, Hammas Zia, Albert Feliciano, Salsabila Laraswari, Muhammad Luthfi, Banita Eka Putri, Agnes Megawati Gultom and Jero Haryono. Thank you to Dwi Adhiasto and Agung Nur Haq for facilitating assistance and Ganjar Cahyadi for assistance during marketplace surveys and bird identification. Thank you to the School of Media and Communication and Biology at the University of Leeds for loaning camera and audio recording equipment during fieldwork.

**Author contribution.** Conceptualisation: S.F., C.H., G.H., T.J., N.H., S.D.; Data collection: S.F., N.H., S.D.; Investigation: S.F., N.H., S.D., C.H., G.H.; Writing – original draft: S.F., C.H., G.H.; Writing – reviewing, critiquing, and editing: C.H., G.H., A.D.

**Financial support.** The Leverhulme Trust funded this research through the Extinction Studies Doctoral Training Program at the School of Biology and Earth and Environment at the University of Leeds. The Society of Conservation Biology Graduate Award fund also provided additional fieldwork funds.

**Competing interest.** The authors declare none.

**Ethics statement.** All work was done with ethical permission from the University of Leeds (AREA FREC 2023-0419-521).

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
