## [Editor Report]

Dear authors,

We are happy to confirm that we would like to include your article in this special issue of Prisms: Extinctions, although both reviewers have requested major revisions, entailing some additional research, but especially adjustments to your argumentation at particular points. Pasted below are the full comments and recommendations. We hope that you will be able to revise your very interesting submission accordingly.

Reviewer 1:

I was very much looking forward to reading this paper and I believe it is correct in identifying a particular “crisis discourse” about Asian songbirds and it is arguably a discourse with a much longer and wider purview than just Asian birds (as the authors also mention towards the end of the paper), the Asian songbirds are a particularly apt example.

I agree with the overall ambition and argument and feel the paper is successful in pointing to a discrepancy between the global conservation discourse and local government and bird-breeder perspectives. At the same time, I feel the paper needs some revision. It is well-written but also at times hard to follow, especially in the analytical sections where it is often not obvious what conclusions the papers draws from the informants' statements. I also think the novelty of the conclusions and recommendations need to be enhanced. I have four overall areas of concern/wonder.

1) the literature on crisis could be expanded and drawn into the discussion also towards the end of the paper

2) The analysis of the informant quotes is not clear and it is not obvious to me that the lumping of informants into “locals” and “internationals” is the most useful. Clearer conclusions are needed from the analysis of the informant/focus group statements.

3) The tendency in the paper to “judge” the truth value of informant quotes by reference to scientific claims feels awkward and reflects perhaps a co-authorship between scholars from ecology/biology and media studies. It speaks to a double audience of ecologists and media studies/human science. The latter in particular might have hard time with the very cursory discussion of concepts used: crisis, discourse, god-like gaze, unequal global structures, etc. The paper sometimes assumes, at times, the same kind of god-like gaze on “discourse” that it accuses the ASC discourse of about “crisis”.

4) The paper ends in a set of conclusions and recommendations that sometimes do not follow from the analysis, that sometimes seems overly optimistic (exchange crisis for challenge and "discourse will disappear?); that raise new issues that are not analysed (such as unequal structures) or that seem very similar to what other have already suggested (cooperation a la Jepson?). I felt instead I needed more insights into what we learnt from the interviews and focus groups interviews.

So I have marked it in need of major revisions, but even smaller adjustment to the language to avoid a clash of epistemologies between the biologists and the media studies people in the author collective, a summary of analytical insights from the interviews (with perhaps more focus on what government agencies or NGOs informants come from); and a revision of the conclusions might be enough.

1) The paper usefully highlights the problems of discourses centred on “crisis”, although the literature on “crisis”n as a specifically discursive phenomenon that the paper challenges is broader than the set of references in the paper. A foundational figure in the critique of the discursive emergence of the term “crisis” is

Koselleck, R. (1998). Critique and Crises: Enlightenment and the Pathogenesis of Modern Society. Massachusetts, The MIT Press.

This gave rise to a broader “anti-crisis” analytical debate about how to align discourses about crisis with “real-world” crisis.

Roitman, J. (2013). Anti-Crisis Durham, Duke University Press.

It is not entirely clear where this paper stands on this issue (even after this issue is addressed at the end of the paper). It is, rightly, critical of the discourse about the Asian Songbird Crisis citing the “contradictory nature of the evidence and the knowledge systems surrounding” it. And it is clearly true that other factors critically contribute to songbird extinctions and the relative impact of trapping and trading may be hard to gauge. But it is not entirely unclear to me where the paper stands on this issue and it sometimes sets up a mismatch between informants and global environmental views that may not be there. For instance:

“An example of how formal declarations of extinction do not necessarily match the situation on the ground is the case of the Javan Pied Starling (Figure 1) (Baveja et al. 2021; Nijman et al. 2021); whilst not formally declared as extinct, it

196 is ‘no longer really in the wild’ (AC_I_1).”

Extinction is here taken as an absolute phenomenon. But does “rolling” or “local” extinction not explain this informant view? Also the term “really” inserted here implies a qualification that is not addressed. Is the lack of “match” here not exaggerated?

2) Informants are marked as either “local”(L) or “international” (I) but it is unclear whether the paper sees any systematic difference between these two groups. Especially in the section “Confirming extinction” the two groups views are represented as equally mixed...

Informants are sometimes cited as having “skewed” discursive views. Indonesian government agencies, for instance, “lacked awareness of this crisis” perhaps due to “insufficient education” (5) while one gets the sense that international informants are in the grips of the “crisis” discourse. The authors of the paper, meanwhile, however above such discursive positions, and they speak from a position of fact, stating apparent truths such as “Substitutions can occur within a species complex, between species in the same genus....” (7), or “A singular crisis, such as the ASC, is often nested within a large one, such as the broader global ecological crisis” (3). This last fact seems to contradict the paper’s abstract and general argument against the use of the term “crisis” because “crises” are not really crises at all but “wicked problems”.

At other times, the authors use their authority to disqualify the truth of informants’s pronouncements: “Although birds evidently play vital roles in tropical forests in Asia, there is a perception from NGOs that these roles are ‘completely unknown and misunderstood by the [birdkeeping] community’ (NG_L_7).” (8)

3) This shift in positions and maintenance of the authors‘ insights into “the reality” of songbird ecology clashes somewhat with the “discourse analysis” approach through which it analyses the statements of its informants. I want to laud the paper again for questioning the “crisis discourse” but an analysis of this discourse also entails reflection upon when the paper itself partakes in this discourse. At the top of page 9 the paper states: “In the context of bird trade, in the long-term, selecting the best singers could lead to forests appearing ’silent' as the loudest species are removed.” But this sentence is itself a reference to the recurrent theme of “silence of bird in the wild” that pervades the ASC discourse and which draws its appeal on that foundational “bird crisis text” by Rachel Carson (1962) called “Silent Spring”. There is in other words a lack of a broader genealogy of the crisis discourse on which ASC draws.

Another feature of the style of the article is that informant pronouncements tend to be directly compared to the environmental-biological literature that has established the ASC in the first place as if they were in direct conversation:

Sometimes this alignment takes the form of validation: “species…[there] is always going to be something else filling it in’ (AC_L_1). This ‘filling in’ of species has been described as a continuous process of ‘rolling local extinctions’ (Jepson and Ladle 2005, see also Jepson 2010).” (7). Here Jepson seems to validate informant AC_L_1.

Sometimes it is framed almost as collegial agreement: "Previously, Harris et al. (2015) hypothesised that persistent demand for certain species in Indonesia’s bird trade indicated low substitutability, meaning that consumers would continue to demand species even as their prices rise, rather than switching to more common (and less expensive) ones. There was also support for this amongst law enforcement; ‘for extinct species the demand in the market will be higher…and the value of

207 the birds will also increase’ (LE_14)". Here it sounds as if LE-14 had read Harris et al.

Then follows an apparent disagreement: "Others from NGOs did not support this interpretation of low substitutability. Several described the process of 'substitution elasticity’ (Fraser et al. 2023), when one species declines, trappers ‘[switch] to

211 another species’ (AC_I_1), seek another bird’ (AC_L_2)". Did AC_I_1 and AC_L_2 in fact quote Harris and Fraser? It almost sounds like it here.

The data is based on sixteen interviews and five focus group interviews with participants from the Indonesian National Police (POLRI), the nature conservation agency (BKSDA), environmental law enforcement (GAKKUM), and agricultural quarantine". The specific institutional position from which the quotes are taken is however unclear, and I wonder whether these position mightn’t explain some of the differences in perception. I suspect members of Polri have much less interest in species, extinctions, and environmental issues generally than members of BKSDA or GAKKUM. With a small number of informants, institutional disaggregation might lead to as interesting differences as a lumping into the super-group “local”.

4) The sense amongst breeders that they 'are helping the government to conserve the birds’ (NG_L_5)“ is an important point highlighted in the paper (9). The paper goes on to suggest that this ”mirrors finding from Jepson et al. (2011) from interviews with Indonesian birdkeepers, who learned that only 13% of participants did ‘not care if species of wild birds go extinct in Indonesia’."

But it this the most relevant follow-on here? Instead of (again) comparing informant statements to scientific articles (see comment on this above), a more relevant analytical follow-up on this would be to trace the discoruse about breeding by the Jokowi government who exactly claims that breeding is conservation. So it is less relevant that breeders of rare birds “believe” that they help the government than to highlight that sections of the Indonesian government itself believes this.

My difficulty in following the arguments of the paper may be related to its interdisciplinary nature. While the authors frequently assume the “god-like” position of knowing whether informants speak the ecological truth, follow the crisis discourse, or lack education and awareness, the text towards the end swings from a ecological truth perspective back into human science constructionism critiquing the “god-like gaze” of the crisis discourse:

"Framing the Southeast Asian bird trade from outside of Indonesia, based on conservation ideologies, resembles Fletcher’s eco-tourist gaze or Haraway’s (1991) ‘god-like gaze’ which grants the ‘viewer’ paternalistic control over events (Lykke 2009). The ASC takes the conservationist gaze, by privileging wildlife conservation over the needs of humans who benefit from songbird trade or even trade actors that are worried about bird declines, exemplified by the Silent Forest Campaign (European Association of Zoos and Aquaria 2023)." (10).

The text then mentions the book Silent Spring and lead on to an alternative framing of ASC around discourses and global inequalities. This is again very laudable but if the claim is that “underlying global structures of inequality ... drive many illegal trades” why are these structures not analysed? If the claim is that “ASC is technically a conservation intervention by the (global) conservation community on Indonesian birdkeepers..(that) has low buy-in from Indonesian government agencies” why is this not analysed in terms of the quotes in the text above and disaggregated so we better understanding what agencies buy in and which do not?

If the argument of the paper is "supporting Jørgensen’s view that ‘extinction [as a process] is not a linear phenomenon’ (Jørgensen 2022), offering a less final and more optimistic perspective. One notable example is the Bali Myna (Figure 1), which ‘[was arguably] saved by captive breeding’ (NG_I_1).“ (11), why is the claim by breeders that they help the government conserve rare birds presented as merely something they ”believe" on page 9?

The paper ends in an excellent conclusion with some important and worthwhile points and recommendations. I agree personally with the ambition but also have some questions:

- Could the paper explain how exchanging the word “crisis” with “challenge” will dismantle the discourse it has identified? What about the underlying structures of equality? Will they be affected by language sanitation of this sort?

- Could the paper explain how its recommendation to enhance “cooperation among governments, conservation practitioners, and birdkeepers” is similar to or different from the same recommendation made by Jepson in 2010?

Jepson, P. (2010). Towards an Indonesian Bird Conservation Ethos: Reflections from a Study of Bird-keeping in the Cities of Java and Bali. Ethno-Ornithology: Birds, Indigenous Peoples, Culture and Society. S. C. Tidemann and A. Gosler. Washington, Earthscan: 313-330.

Reviewer 2:

The central theoretical proposal of this article – the problematisation of term “crisis” as applied to extinction – is an interesting and intriguing one, and I read it with interest. I find the analysis of the complexities of extinction discourses in the section “Confirming extinctions” particularly interesting, and worth expanding on. There are, however, issues with the paper that call for major revisions in order to really flesh out and solidify the argument.

I cannot comment much on the methods used for this research, as I am not trained in ethnographic methods. My main criticism, however, has to do with the choice of respondents, and pertains to the section “Response of breeders to trade-induced declines”. The title is somewhat misleading, as the respondents are entirely comprised of actors from the academic sector, law enforcement, and NGOs. The breeders and birdkeepers are entirely absent from this section and only presented through the lens of the respondents’ opinions about them. This one-sided choice represents a troubling bias in the research and needs to be either addressed frontally and explained or supplemented more robustly.

This weakness means that it is not quite clear why “the emphasis by law enforcement on ecosystem-level impacts may offer a more useful route for valuing songbirds for their ecological roles than through a conservation crisis”. While this is certainly a compelling hypothesis, especially if grounded in a decolonial analysis of the socio-economic factors that underpin bird trade in Indonesia, the paper does ultimately not provide a solid argument for why the phrase ‘Asian Songbird Challenge’ would work better in bringing a variety of actors together.

Overall, I find that the article speeds through many of its points in a way that undermines the strength of the concluding proposal, and I recommend fleshing it out significantly. One symptomatic tendency in the text is, in my opinion, the use of humanities scholarship sprinkled throughout. The references often feel anecdotal and like afterthoughts, I particular in the sections quoting Haraway (her “god-trick” is more about the construction of objectivity in experimental science than it is about colonialism) and Jørgensen. The use of humanities scholarship when it comes to decolonial issues is more successful, and I would like to see the others woven into the argument more solidly as well.

---

## [Editor Report]

Whilst both reviewers praise the revised version of this article, with one recommending publication with no further revisions, the other has called for some further minor revisions. Could you please give consideration as to how you might be able to respond to the following recommendations in particular:

1. The paragraph “Crises can be temporary conditions…” (39) would benefit from being much more fleshed out, as it speeds through a very condensed presentation of several conflicting definitions of the term in a way that I find insufficiently rigorous for a journal article.

2. The sentence “For example, indigenous groups like the Anishinaabeg people (Great Lakes, North America) observe that animals actively withdraw from the world when societies break human-animal treaties (O’Key 2023).” (54-55) should be expanded on to enrich the preceding paragraph, as it feels like a slightly random and decorative example added to the argument.

3. Paragraph “Looking at environmental humanities scholarship…”: I would like to see both the question of extinction as non-linear and reversible and the comparison to recovering stocks in commercial fishing fleshed out. This is a very interesting point, but the paragraph does not do very much with the participant it quotes here. As a result, the paragraph floats oddly at the end of this section, when it could be tethered to the rest of the argument more strongly.